# Metformin Induces Resistance of Cancer Cells to the Proteasome Inhibitor Bortezomib

**DOI:** 10.3390/biom12060756

**Published:** 2022-05-28

**Authors:** Camille Schlesser, Thomas Meul, Georgios Stathopoulos, Silke Meiners

**Affiliations:** 1Comprehensive Pneumology Center (CPC), Helmholtz Center Munich, Ludwig-Maximilians University, Max-Lebsche Platz 31, 81377 Munich, Germany; schlesser.camille23@gmail.com (C.S.); thomas.meul@gmx.de (T.M.); stathopoulos@helmholtz-muenchen.de (G.S.); 2Member of the German Center for Lung Research (DZL), 35392 Gießen, Germany; 3Laboratory for Molecular Respiratory Carcinogenesis, Department of Physiology, Faculty of Medicine, University of Patras, 26504 Rio, Greece; 4Research Center Borstel/Leibniz Lung Center, Parkallee 1-40, 23845 Borstel, Germany; 5Airway Research Center North (ARCN), German Center for Lung Research (DZL), 23845 Sülfeld, Germany; 6Institute of Experimental Medicine, Christian-Albrechts University Kiel, 24118 Kiel, Germany

**Keywords:** metformin, proteasome inhibitor, resistance, 26S proteasome, proteasome activity

## Abstract

The anti-diabetic drug metformin is currently tested for the treatment of hematological and solid cancers. Proteasome inhibitors, e.g., Bortezomib, are approved for the treatment of multiple myeloma and mantle cell lymphoma but are also studied for lung cancer therapy. We here analyzed the interaction of the two drugs in two cell lines, namely the mantle cell lymphoma Jeko-1 and the non-small-cell lung cancer (NSCLC) H1299 cells, using proliferation and survival assays, native-gel analysis for proteasome activity and assembly, and expression analysis of proteasome assembly factors. Our results demonstrate that metformin treatment induces resistance of cancer cells to the proteasome inhibitor Bortezomib by impairing the activity and assembly of the 26S proteasome complexes. These effects of metformin on proteasome inhibitor sensitivity in cancer cells are of potential relevance for patients that receive proteasome inhibitor therapy.

## 1. Introduction

Metformin is a safe and potent antihyperglycemic drug widely prescribed for the treatment of type 2 diabetes mellitus [1]. Mechanistically, metformin activates AMP-activated kinase (AMPK), a key sensor of the energetic state of the cell that switches off energy-consuming pathways, including mammalian target of rapamycin (mTOR) signaling [2] via inhibiting the lysosomal proton pump v-ATPase [1,3,4]. At higher doses, metformin inhibits respiratory chain complex I of mitochondria [5,6]. Besides its antihyperglycemic potential, metformin treatment reduces the risk of diabetic patients developing cancer and improves their prognosis upon cancer diagnosis [7,8,9]. These observations have contributed to an increasing interest in metformin as a potential antitumorigenic agent. Hence, its antineoplastic effects are currently studied in several clinical trials, including those on hematologic and lung tumors (see Appendix A for an overview of selected clinical studies).

Cancer cells rely on the effective turnover of proteins via the ubiquitin-proteasome system [10]. Ubiquitinated proteins are degraded by the 26S proteasome, which is composed of a 20S catalytic core and one or two 19S regulators that bind to the symmetric ends of the 20S particle [11]. The 19S regulator binds, deubiquitinates, and unfolds the substrate so that it can be channeled into the 20S for degradation. The unfolded protein chain is then hydrolyzed by the three active sites of the 20S proteasome into small peptides. These peptides are further degraded to recycle amino acids for protein synthesis. Proteasomal degradation is thereby closely coupled to protein synthesis [12]. Inhibition of the catalytic active sites of the proteasome with specific proteasome inhibitors blocks cellular proliferation and induces apoptosis, thus effectively reducing cancer growth and tumor spread [13,14]. The proteasome inhibitor Bortezomib (Velcade^®^) received first approval by the US Food and Drug Administration (FDA) in 2003 as a single agent for the treatment of patients with relapsed or refractory multiple myeloma [15]. To date, three different catalytic proteasome inhibitors are approved as a first-line treatment option for patients with newly diagnosed multiple myeloma (MM) or mantle cell lymphoma (MCL) [16]. Proteasome inhibitors are also tested as treatment options for solid cancers such as lung cancer (Available online: https://www.clinicaltrial.gov, accessed 13 May 2022 and Appendix A).

We previously demonstrated that high doses of metformin conferred increased resistance toward the proteasome inhibitor Bortezomib in primary mouse fibroblasts [17]. Here, we demonstrate that metformin also confers increased resistance to Bortezomib in cancer cells. These findings might have important therapeutic implications for patients with multiple myeloma or mantle cell lymphoma that receive proteasome inhibitor and metformin co-treatment as currently tested in clinical trials (Available online: https://www.clinicaltrial.gov, accessed 13 May 2022 and Appendix A).

## 2. Materials and Methods

### 2.1. Cell Culture and Treatments

Jeko-1 (ATCC-Nr. CRL-3006) and H1299 (ATCC-Nr.CRL-5803) cancer cells were cultured using RPMI-1640 medium supplemented with 10% fetal bovine serum (F0804, Sigma Aldrich, Taufkirchen, Germany) and 100 U/mL penicillin/streptomycin (Thermo Fisher Scientific, Bremen, Germany). For treatments, cells were plated onto 6-/24- or 96-well plates (depending on the respective analysis method) and incubated overnight. The attached cells were then washed with PBS, and the culture medium was changed to the treatment medium as indicated in the figures and respective legends. For Jeko-1 suspension cells, the washing step with PBS was omitted, and the treatment medium was directly added to the cells. The total duration of metformin treatment was 72 h. Sensitivity of the different cell lines toward bortezomib was assessed upon treatment of cells for 24 h with different doses of bortezomib on the day following plating without medium exchange. Bortezomib treatments in combination with metformin pre-treatment were performed by addition of bortezomib following 24 or 48 h of pre-treatment. The duration of bortezomib treatment was 24 or 48 h, depending on the cell line. Bortezomib doses used for the respective experiments are indicated in the figure legends.

### 2.2. Cell Proliferation Assay

Cellular proliferation rates (doublings per day) were assessed as published by Sullivan et al. [18] using the following formula:Proliferation rate (doublings per day) = log_2_ (Final cell count (day 4)/Initial cell count (day 0))/4 (days)

### 2.3. MTT Metabolic Activity Assay

Cell viability was assessed by measurement of metabolic activity using the 2,5-diphenyltetrazolium bromide (MTT) assay according to manufacturers’ instructions (Sigma Aldrich, Taufkirchen, Germany). Cells were seeded into round-bottom 96-well plates using a normal cell culture medium and incubated overnight. The next day, the medium was changed to the respective treatment or control medium as indicated in the figure legends. Bortezomib was added after 24 or 48 h of pre-treatment. Following 72 h of total treatment time, freshly prepared thiazol blue tetrazolium bromide solution (5 mg/mL PBS) and cells were incubated at 37 °C for one hour before MTT absorbance measurement at 570 nm using a Sunrise plate reader (Tecan, Life Sciences, Männedorf, Schitzerland).

### 2.4. Live/Dead Assay Using Annexin V/PI

Apoptosis or necrosis of the cells was assessed by staining cells with an Annexin V antibody and propidium iodide (PI). The attached cells were seeded into 6-well plates and incubated overnight. After washing cells with PBS the next day, the medium was exchanged for treatment or control medium. Bortezomib was added as indicated. Following treatment, cells were washed with PBS and harvested by trypsinization and resuspension in the medium. Suspension cells were seeded into 6-well plates and incubated overnight. The next day, treatment or control medium were added to the cells. Following treatment, cells were harvested, washed twice with PBS, and resuspended in Annexin V binding buffer (10 mM HEPES, 150 mM NaCl, 2.5 mM CaCl_2_, pH 7.4). Then, 5 µL Anti-Annexin V FITC (BD Biosciences, Heidelberg, Germany) and 10 µL of PI staining solution (BD Biosciences, San Jose, CA, USA) were added to the cells for 15 min at room temperature in the dark before the addition of Annexin V binding buffer. Samples were kept on ice and measured by flow cytometry using Becton Dickinson LSRII (Becton Dickinson, Heidelberg, Germany) and analyzed using FlowJo software (Version 10, Becton Dickinson, Heidelberg, Germany).

### 2.5. Protein Extraction

Extraction of protein under preservation of their native structure was performed using TSDG lysis buffer (10 mM Tris/HCl, 10 mM NaCl, 1.1 mM MgCl_2_, 0.1 mM EDTA, 1 mM DTT, 1 mM NaN_3_, 10% (*v*/*v*) glycerol, pH 7.0). TSDG lysis was performed by resuspending cell pellets in 40–200 µL (depending on the pellet size) of TSDG lysis buffer supplemented with 1x cOmplete^®^ protease inhibitor cocktail (Roche Diagnostics, Basel, Switzerland) and 1x PhosSTOP (Roche Diagnostics). Cell debris was removed by 20 min of centrifugation at 14, 000 rpm at 4 °C. The supernatant containing the protein lysate was collected.

### 2.6. Bicinchoninic Acid Assay—BCA Assay

Protein concentrations of extracted lysates were determined directly after lysis using the Pierce BCA protein assay kit according to the manufacturer’s instructions (Thermo Fisher Scientific, Bremen, Germany), and absorbance was measured at a wavelength of 562 nm using the Sunrise Plate Reader (Tecan, Life Sciences, Männedorf, Schitzerland).

### 2.7. Western Blot Analysis

Western blot analysis was performed as previously described [17] using native TSDG extracts and the antibodies summarized in Table 1. Quantification of the detected bands was performed using ImageLab software (Biorad, Hercules, CA, USA). Full Western blot and quantification data are summarized in Appendix A. 

### 2.8. Native PAGE Proteasome Analysis

Native gel analysis was performed according to our previously published protocol using native TSDG protein extracts and precast 3–8% gradient NuPAGE Novex Tris-acetate gels (Life Technologies, Olber-Olm, Germany) [19]. For in-gel activity assay, native gels were incubated for 30 min in reaction buffer containing 50 mM Tris, 1 mM ATP, 10 mM MgCl_2_, 1 mM DTT, and 0.05 mM Suc-LLVY-AMC (for analysis of the chymotrypsin-like activity of the proteasome). Before blotting, gels were incubated for 15 min in solubilization buffer, then blotted onto a PVDF membrane using standard Western blot techniques as previously described [19].

### 2.9. Statistical Analysis

Data are shown as mean ± SEM. Statistical analysis was performed using Graph Pad Prism software (Version 9, San Diego, CA, USA). Applied tests for statistical analysis are indicated in each figure legend. Statistical analysis of MTT assays comparing viability of metformin pre-treated to non-pre-treated cancer cells upon bortezomib exposure was performed using 2-way ANOVA with Bonferroni multiple comparisons test. One-way ANOVA was used to determine the statistical significance of quantified Annexin V/PI stainings. Student’s one-sample t-test was used to determine significant differences in protein expression and proteasome complex levels, as determined by Western blot and native gel analysis, of metformin treated compared to control cells. Significance is indicated in the figures as * *p* < 0.05; ** *p* < 0.01; *** *p* < 0.001.

## 3. Results

We here analyzed the effect of metformin on proteasome inhibitor sensitivity in human cancer cells. Specifically, we analyzed two distinct tumor cell lines, i.e., the mantle cell lymphoma cell line Jeko-1 and H1299 cells, that are derived from solid NSCLC tumors.

### 3.1. Metformin Increases Resistance towards the Proteasome Inhibitor Bortezomib

We first confirmed the antiproliferative effects of metformin in the two cancer cell lines. The proliferation rate of both Jeko-1 and H1299 cells was clearly reduced by treatment with 10 mM metformin for 72 h (Figure 1a). Next, we tested whether metformin treatment affects cell viability in response to proteasome inhibitor treatment. For that, cells were pre-treated with or without metformin for 48 h, exposed to increasing doses of Bortezomib (5–50 nM for Jeko-1 cells; 5–100 nM for H1299 cells) for 24 or 48 h, and cell viability was determined via MTT assays. We noticed that Jeko-1 cells were more sensitive toward Bortezomib-induced cell death compared to H1299 cells and therefore adjusted Bortezomib treatment time to 24 h. Of note, metformin reduced the sensitivity of both Jeko-1 and H1299 cells towards Bortezomib treatment and conferred resistance towards proteasome inhibition (Figure 1b). While Bortezomib reduced cell viability by about 80% in the mantle cell lymphoma cell line Jeko-1 at a dose of 35 nM, metformin pre-treated cells maintained 65% of cell survival even at the maximal dose of 50 nM Bortezomib. This effect was even more pronounced in the lung cancer cell line H1299 with a 90% reduction in cell survival with 25 nM Bortezomib but about 60% survival even at the highest dose of 100 nM Bortezomib without metformin pre-treatment. We confirmed this finding by quantifying cell death using flow-cytometry-based annexin V detection. Pre-treating Jeko-1 cells with metformin completely abrogated the induction of apoptosis induced by 20 nM Bortezomib and strongly reduced necrotic cell death (Figure 1c). Metformin similarly increased the resistance to Bortezomib-induced apoptotic cell death in H1299 cells (Figure 1d).

### 3.2. Metformin Impairs Assembly and Activity of 26S Proteasomes in Cancer Cells

We next investigated the mechanism that may cause metformin-induced Bortezomib resistance. Based on our previous study, where we had demonstrated that metformin-induced metabolic remodeling impairs assembly and activity of 26S proteasome complexes by transcriptional regulation of assembly factors [17], we here analyzed 26S proteasome assembly in detail. The activity and amounts of different proteasome complexes were analyzed in native cell lysates by native gel electrophoresis followed by chymotrypsin-like (CT-L) substrate overlay assay and immunoblotting for 20S α1-7 subunits [19]. Metformin treatment of both Jeko-1 and H1299 cell lines significantly reduced the activity of singly capped 26S and doubly capped 30S complexes. In Jeko-1 cells, 20S activity was increased (Figure 2a). Immunoblotting of the native gels for 20S subunits revealed diminished amounts of 26S and 30S complexes in metformin-treated cells, while the amount of the 20S complexes was increased (Figure 2a). This indicates either disassembly of 26S complexes or impaired assembly of preexisting 20S with 19S regulators. To further investigate potential defects in the assembly of the 26S proteasome upon metformin treatment, we analyzed the expression of several 26S proteasome assembly factors, such as Rpn6, p28, and S5b, as well as components of the 20S proteasome by Western blotting. Of note, metformin treatment upregulated expression of only S5b in Jeko-1 cells, while the assembly chaperone p28 was significantly downregulated in H1299 cells (Figure 2b). Rpn6 expression was not altered. While S5b functions as an inhibitor, p28 acts as a chaperone of 26S proteasome assembly [20,21]. Accordingly, these data indicate that metformin treatment impairs the assembly of 26S proteasome complexes in Jeko-1 and H1299 cancer cells by differentially regulating assembly factors. Of note, expression data for Jeko-1 and H1299 cell lines obtained from publicly available GEO data sets indicate differential expression of proteasome assembly factors at baseline, with S5b being significantly lower expressed in Jeko-1 compared to H1299 cells (data not shown), which may add to the differential metformin-induced regulation of S5b and p28 in the two cell lines. We also analyzed the activation of the p70 ribosomal S6 kinase, a key enzyme involved in the activation of protein synthesis via the mTOR pathway [22], as we had previously observed that metabolic regulation of 26S activity involved specific regulation of proteasome assembly factors via the mTORC1 pathway [17]. While Jeko-1 cells did not show altered p70 S6 kinase activation, both, expression of the total and phosphorylated forms of p70 S6 kinase, were significantly reduced in H1299 cells (Figure 2b). These results further add to the notion that the two cancer cell lines of hematopoietic (Jeko-1) versus lung cell origin (H1299) use different mechanisms to regulate 26S proteasome activity upon metformin treatment.

## 4. Discussion

We here observe increased resistance of metformin-treated tumor cells toward Bortezomib. It is important to note that we applied suprapharmacological doses of metformin which are well known to block respiratory complex I activity [6]. According to our previous findings on reduced proteasome inhibitor sensitivity in complex I mutant respiration-deficient cells [17], the inhibitory effect of metformin on mitochondrial respiratory chain function seems crucial for increasing the resistance toward proteasome inhibitors. This reduced sensitivity towards proteasome inhibitor treatment might provide an explanation for the previously reported observation that metformin has protective effects on proteasome-inhibitor-induced cardio- and vascular toxicity in mice [23,24,25]. In contrast to our study, other reports demonstrated additive cytotoxic effects of metformin and proteasome inhibitors on multiple myeloma (MM) cells [26,27]. In these studies, metformin suppressed proteasome-inhibitor-induced autophagy and promoted apoptotic cell death. It is well feasible that the different doses and cell types used in these studies account for the different results.

Our mechanistic data suggest that metformin treatment impairs the assembly of 26S proteasome complexes in Jeko-1 and H1299 cancer cells, possibly by differentially regulating assembly factors. While p27, p28, and S5b act as regulatory particle assembly chaperones, so-called RACs [28], Rpn6 has been described as a rate-limiting subunit of the 26S proteasome, which regulates assembly of the 26S complex either by its expression level [13,14] or its phosphorylation status [15]. Our findings are fully in line with our previous findings, where we observed transcriptional regulation of assembly factors upon respiratory complex I inhibition [17]. The only minor regulation of single assembly factors in Jeko-1 and H1299 cells, however, suggests that additional—currently unknown—factors might be involved in hindering the assembly of 26S proteasome complexes upon metformin treatment in cancer cells. When analyzing cBioPortal (available online: http://www.cbioportal.org, accessed on 23 May 2022) for the mutational status of the assembly factors in Jeko-1 and H1299 cells, we noticed that Jeko-1 cells contain an amplification of the S5b-encoding gene PSMD5 while H1299 cells have a deletion of Rpn6 (PSMD11). Mutations in assembly factors were detected in 11% of all the 1806 cancer cell lines in this database, suggesting an additional and currently unrecognized genetic regulation of 26S proteasome assembly in cancer cells. Along this line, it is worth mentioning that we were unable to reverse the increased metformin-induced resistance to proteasome inhibition by supplementation with the alternative electron acceptor pyruvate (data not shown). Pyruvate or aspartate-mediated restoration of 26S proteasome assembly was demonstrated by us previously in primary cells and was related to mTORC1-mediated activation of 26S assembly factors [17]. This thus indicates that cancer cells have adopted complex metabolic changes compared to primary cells, which influence 26S proteasome function and inhibitor resistance, as also suggested by Peter Tsvetkov’s work [29,30]. In accordance with our data shown here, previous studies reported resistance to proteasome inhibition due to reduced expression of 19S subunits, which resulted in defective 26S proteasome assembly and activity [31,32,33].

Taken together, treatment of hematologic or solid cancers with metformin might favor resistance to proteasome inhibitor co-treatment. High concentrations of metformin might be obtained locally in tumor cells due to increased expression of metformin importing organic cation transporters (OCTs) by cancer cells resulting in the local accumulation of metformin even at low plasma concentrations [34,35]. Accordingly, the here described effects of metformin on proteasome inhibitor sensitivity in cancer cells are of potential relevance for patients that receive proteasome inhibitor therapy. Therefore, although metformin seemingly has numerous beneficial effects in both diabetic and/or cancer patients, a more profound study of its impact on the effectiveness of proteasome-inhibitor-based treatments in cancer is of high clinical interest.

## Figures and Tables

**Figure 1 biomolecules-12-00756-f001:**
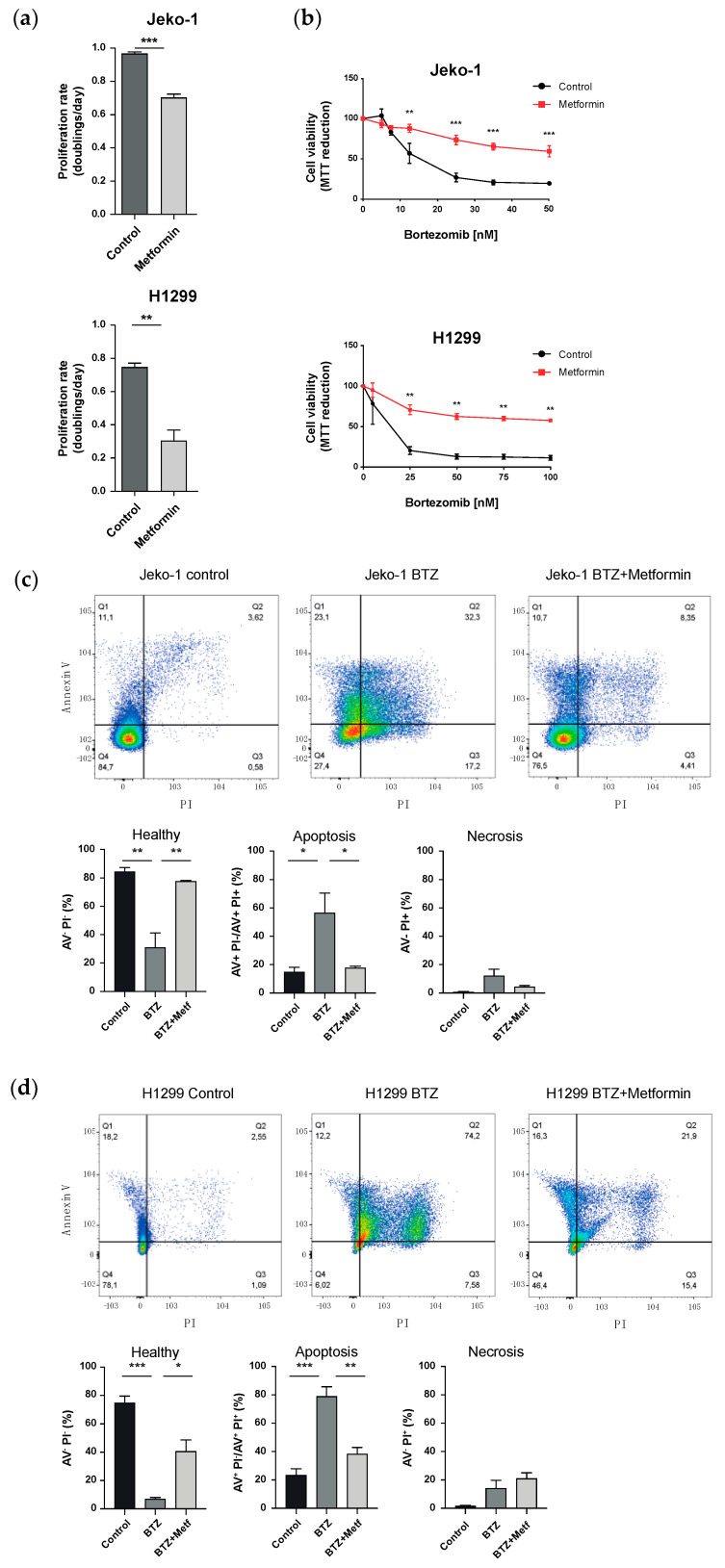
Metformin reduces cancer cell proliferation but increases resistance toward proteasome inhibition. (**a**) Proliferation rates of Jeko-1 and H1299 cells treated with 10 mM metformin for 72 h. Proliferation rates were determined as described in the methods section. Bar graphs show mean + SEM (*n* = 3 independent experiments per cell line). Significance was determined using Student’s paired t-test. (**b**) MTT metabolic activity assays for the assessment of cellular viability of Jeko-1 and H1299 cells pre-treated with either control or 10 mM metformin medium for 48 h (Jeko-1) or 24 h (H1299), followed by 24-hour (Jeko-1) or 48-hour (H1299) treatment with increasing doses of bortezomib. All values were normalized to the 0 nM bortezomib value of the specific cell line and treatment group and are displayed as mean +/− SEM. Significance was determined using two-way ANOVA with Bonferroni multiple comparison test. (**c**) Representative flow cytometry analysis and quantification of Annexin V/PI stained Jeko-1 cells pre-treated with 10 mM metformin for 48 h, followed by 20 nM bortezomib treatment for 24 h. (**d**) Representative flow cytometry analysis and quantification of Annexin V/PI stained H1299 cells pre-treated with 10 mM metformin for 24 h, followed by 75 nM bortezomib treatment for 48 h. (**c**,**d**) Quantifications show percentages of healthy (Annexin V^−^/PI^−^), early (Annexin V^+^/PI^−^), or late apoptotic (Annexin V^+^/PI^+^), and necrotic (Annexin V^−^/PI^+^) cells. Bar graphs show mean + SEM (*n* = 3 independent experiments). Significance was determined using one-way ANOVA. * *p* < 0.05; ** *p* < 0.01; *** *p* < 0.001.

**Figure 2 biomolecules-12-00756-f002:**
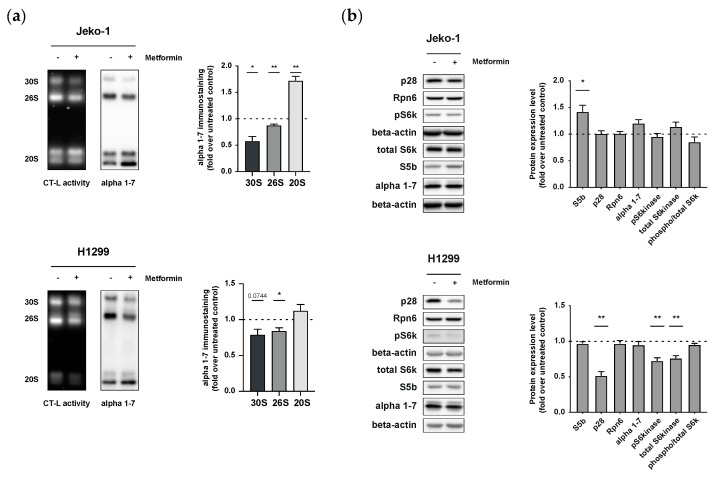
Metformin treatment leads to downregulation of 30S and 26S proteasome assembly. (**a**) Representative in-gel chymotrypsin-like substrate activity assay of native cell lysates from Jeko-1 and H1299 cells treated with 10 mM metformin for 72 h (left panels) followed by immunostaining of 20S alpha 1-7 subunits of blotted native gels (right panel). Quantification of 30S, 26S, and 20S is presented as mean + SEM relative to the untreated control. Significance was determined using Student’s one-sample t-test. (**b**) Representative Western blot analysis of 26S assembly factors (S5b, p28, and Rpn6) and 20S (alpha 1-7 subunit) as well as of phosphorylated and total p70 S6 kinase (S6k) expression in Jeko-1 and H1299 cells treated with 10 mM metformin for 72 h. Beta-actin was used as loading control. Bar graphs of densitometric analysis show mean + SEM relative to the untreated control. Significance was determined using Student’s one-sample t-test. * *p* < 0.05; ** *p* < 0.01.

**Table 1 biomolecules-12-00756-t001:** Antibodies used in this study.

Antigen	Dilution	Product No (Manufacturer)
P28 (PSMD10)	1:1000	ab182576 (Abcam, Cambridge, UK)
Rpn6 (PSMD11)	1:1000	NBP1-46191 (Novus Biologicals, Biotechne, Wiesbaden, Germany)
S5b (PSMD5)	1:1000	ab137733 (Abcam, Cambridge, UK)
Alpha 1-7 (MCP231)	1:1000	Ab22674 (Abcam, Cambridge, UK)
Phospho-S6 kinase (Ser371)	1:1000	9208 (Cell signaling, Danvers, MA, USA)
S6 kinase	1:1000	2708 (Cell Signaling, Danvers, MA, USA)
HRP conjugated anti-mouse IgG	1:40,000	7076 (Cell signaling, Danvers, MA, USA)
HRP conjugated anti-mouse IgG	1:40,000	7074 (Cell signaling, Danvers, MA, USA)

## Data Availability

Not applicable.

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
