# Peer review of "Metformin Induces Resistance of Cancer Cells to the Proteasome Inhibitor Bortezomib"

_biomolecules, 2022, doi:10.3390/biom12060756_

Round 1

Reviewer 1 Report

Schlesser et al reported here that high-dose metformin could cause bortezomib resistance in cancer cells by downregulating proteasome assembly and activity. This is a continuation of their previous work showing a similar response in MEFs. In the current study, the authors found in two cancer cell lines (Jeko-1 and H1299) that metformin alone led to reduced cell proliferation, whereas co-treatment of cells with metformin and bortezomib ameliorated the pro-apoptotic effect of the latter drug. This phenomenon was attributed to decreased proteasome activity (of the 26S/30S holoenzymes) after metformin treatment, which caused dysregulation of proteasome assembly chaperones such as S5b and p28.

There are 3 major concerns about this work.

  1. The authors simply repeated a small portion of experiments in their 2020 Cell Rep. paper, using different cell lines. Although they tried to make a connection with cancer treatment, conceptually there seems to be no significant advancement towards a deeper understanding of proteasome regulation.
  2. Many metformin-related papers, including this one, relied on very high concentrations of the drug to achieve mitochondrial inhibition. The results are likely not physiologically relevant, esp. considering the recent discovery of a true metformin target (PEN2, Ma et al, Nature, 2022) which responds to micromolar metformin (as opposed to 10 mM used in this study). Admittedly, metformin is a fairly safe drug and may occasionally accumulate to high concentrations in the body. However, given our renewed view of metformin action, the authors can at least try using a lower concentration of the drug or test other mito-targeting compounds at relevant dosages.
  3. In their Cell Rep. paper, the authors identified mTOR-regulated protein translation control as a potential mechanism for the downregulation of proteasome chaperones in response to metformin. But in the current study they did not show if the same mechanism was at work in the cancer cells, nor did they explain why S5b and p28 were regulated differently in each cell line. Moreover, the rather acute responses seen here are unlikely to be explained by the change of cell state/identity as proposed by Tsvetkov et al.

Minor points:

Line 164: “The doubling time of both Jeko-1 and H1299 cells was clearly reduced by treatment with 164 10 mM metformin” – The doubling time should be INCREASED as metformin inhibits cell proliferation.

Figure 1c, d: Axis labels of the FACS data are illegible.

Lines 251-255: Despite its structural role in proteasome assembly, Rpn6 is not considered as an assembly factor. In addition, S5b should not be labeled as an inhibitor, as the base chaperones are all important for assembly, even though all of them can block RP-CP interaction when expressed in excess.

Reviewer 2 Report

The manuscript by Schlesser et al. “Metformin induces resistance of cancer cells to the proteasome inhibitor Bortezomib” describes resistance of cancer cells treated with well-known anti-diabetic drug to proteasome inhibitor Bortezomib. The manuscript is clear, well-written and describes an intriguing observation. I suppose that the manuscript will be interesting to researchers working in the field, and I think it could be published in Biomolecules after a Minor revision.

I have only few comments:

Major point

-Does the Metformin alone have an effect in the MTT test or affect Annexin or PI staining of cells? Perhaps this additional control might be worth to add.

Minor points

-Please give a link or a number for trials that are testing the specific anti-tumor effect of metformin or PIs instead of providing a general link “www.clinicaltrials.gov”

-Lane 164. It is written that “The doubling time of both Jeko-1 and H1299 cells was clearly reduced by treatment with 10 mM metformin”. However, if we consider the doubling time as the time needed to double the cell population in the culture it was obviously increased e.g. cells started to grow slowly as it could be seen in the Figure 1a where the number of doublings per day is significantly decreased for the metformin-treated cells.

-Please increase the font size of the axis labels Figure 1c,d (flow cytometry)

Finally, is there an activity increase of 11S capped proteasomes following the metformin treatment in Jeko1 cells? I do not insist on any corrections rather a question of curiosity.

P.S. I liked the quality of Native PAGE very much!

Round 2

Reviewer 1 Report

The authors have provided reasonable replies to my concerns.